# All-dielectric magnetic metasurface for advanced light control in dual polarizations combined with high-Q resonances

Daria O. Ignatyeva [1,2,3✉], Dolendra Karki [4], Andrey A. Voronov [1,3], Mikhail A. Kozhaev[2,3,5], Denis M. Krichevsky [2,3,6], Alexander I. Chernov [3,6], Miguel Levy[4] & Vladimir I. Belotelov [1,2,3]

Nanostructured magnetic materials provide an efficient tool for light manipulation on sub-nanosecond and sub-micron scales, and allow for the observation of the novel effects which are fundamentally impossible in smooth films. For many cases of practical importance, it is vital to observe the magneto-optical intensity modulation in a dual-polarization regime. However, the nanostructures reported on up to date usually utilize a transverse Kerr effect and thus provide light modulation only for p-polarized light. We present a concept of a transparent magnetic metasurface to solve this problem, and demonstrate a novel mechanism for magneto-optical modulation. A 2D array of bismuth-substituted iron-garnet nanopillars on an ultrathin iron-garnet slab forms a metasurface supporting quasi-waveguide mode excitation. In contrast to plasmonic structures, the all-dielectric magnetic metasurface is shown to exhibit much higher transparency and superior quality-factor resonances, followed by a multifold increase in light intensity modulation. The existence of a wide variety of excited mode types allows for advanced light control: transmittance of both p- and s-polarized illumination becomes sensitive to the medium magnetization, something that is fundamentally impossible in smooth magnetic films. The proposed metasurface is very promising for sensing, magnetometry and light modulation applications.

[1] Faculty of Physics, Lomonosov Moscow State University, Moscow, Russia. [2] Crimean Federal University, Simferopol, Russia. [3] Russian Quantum Center, Moscow, Russia. [4] Physics Department, Michigan Technological University, Houghton, MI, USA. [5] Prokhorov General Physics Institute of the Russian Academy of Sciences, Moscow, Russia. [6] Center for Photonics and 2D Materials, Moscow Institute of Physics and Technology (National Research University), Dolgoprudny, Russia. ✉email: ignatyeva@physics.msu.ru

Light modulation by magnetic field has been successfully used for various practical applications such as magneto-optical modulators, isolators, routers and other non-reciprocal devices[1–8], magneto-optical biosensors[9–14] as well as magnetometers[15–18]. For all of the named areas, it is important, on the one hand, to increase the magneto-optical performance of the nanostructure, and on the other hand, to achieve miniaturization of the device and make it compatible with integrated optics. Apart from that, in many cases it might be advantageous to modulate both orthogonal linear polarizations, i.e., p- and s-polarizations of light. For example, a simultaneous control of two polarizations would allow the magneto-optical biosensing in a dual-channel regime where one polarization is utilized for the measurement of the bulk, and the other –of surface characteristics[19,20] and would significantly improve vector magnetometers based on the magnetooptical measurements in orthogonal polarizations for the detection of the in-plane magnetic field components[21,22].

Usually, the transverse magnetooptical Kerr effect (TMOKE) which modulates the intensity of the light reflected from a magnetic film depending on the magnetization direction[1] is among the main tools for such applications. Though TMOKE does not exceed hundredths of percent for a smooth transparent magnetic film, it could be significantly enhanced in various nanostructured materials and metasurfaces exhibiting optical mode excitations[23–30]. It is worth noting that propagating modes (such as guided or surface plasmon-polariton modes) are more promising, since their wavevector is non-reciprocally modified by the external magnetic field[31] in contrast to the localized modes (localized plasmons or Mie modes) in different kinds of nanoantennas, demonstrating a very moderate TMOKE increase[32,33] although both of them exhibit strong light localization inside the nanostructure[34,35]. Thus, magnetoplasmonic crystals consisting of a periodic 1D or 2D metallic grating and a magnetic film are one of the most efficient nanostructures providing a significant enhancement of the light modulation both in reflected and transmitted light[14,36–38]. However, it is important that observation of the magneto-optical effects in magnetoplasmonic structures is accompanied by a significant decrease of the base signal due to high absorption in metals, also leading to resonance broadening. Moreover, TMOKE in the plasmonic structures considered so far, modulates only p-polarized light leaving the s-polarized component unaffected. At the same time, all-dielectric nanostructures[39] are promising in the sense that resonance high quality factors and transparency might be combined with high magnetooptical modulation efficiencies. The 1D[40,41] and 2D[42] all-dielectric magnetic gratings and

nanoparticles[43] were numerically predicted to increase some magneto-optical effects. However, fabrication of such complex shapes is challenging, and, on the other hand, the resulting optical response would be significantly affected not only by the material losses, but also by the surface roughness and fabrication inaccuracies that are neglected in these theoretical studies[40–43]. Although recent experimental studies[44] reveal the enhancement of TMOKE in 1D all-dielectric gratings, it is still observed only for p-polarized illumination and completely vanishes for s-polarized light. Experimentally, up to now all-dielectric magnetic nanostructures have been implemented in the form of individual nanodisks only for ferromagnetic resonance measurements[45]. The magneto-optical properties of nanopatterned dielectrics have not been studied yet.

Here we fabricate and investigate a magnetophotonic structure —a magnetic metasurface made of 2D iron-garnet subwavelength nanopillar arrays on a thin iron-garnet film fabricated by electron-beam lithography. In contrast to magnetoplasmonic structures, where the transmission coefficient $T$ usually does not exceed the value of about several percent, in this presented magnetic metasurface, the ratio between transmission $T$ and reflection $R$ could be tuned[46], making it possible to enhance $T$ and the magneto-optical response simultaneously. The fabricated magnetic metasurface has a transmittance of more than 50%, that is almost unreachable with conventional plasmonic structures. A set of propagating quasi-waveguide modes is excited in the structure causing a multifold enhancement of the light intensity modulation via the magnetic field compared to the smooth iron-garnet film. The prominent feature of the structure is the possibility to observe the transversal intensity magneto-optical effect in transmission not only for p-, but also for s-polarization of the incoming light. Observation of the enhancement of the magneto-optical intensity effect for both incident polarizations is also a unique feature of the proposed magnetic metasurface totally absent for a smooth magnetic film.

## Results

**Optical modes of 2D magnetic metasurface.** The all-dielectric magnetic metasurface was fabricated by electron-beam lithography on a 300-nm thick Bi-substituted iron-garnet film $(Bi_{0.7}Gd_{0.3}Lu_{2.0}Ga_{0.8}Fe_{4.2}O_{12})$ (BIG) epitaxially grown on a gadolinium gallium garnet (GGG) substrate (see Methods). Substitution of rare-earth ions with Bi significantly enhances the magneto-optical response[47]. The film can be fully magnetized in-plane by magnetic field of $H_{in} = 30$ Oe. Nanopillars of diameter

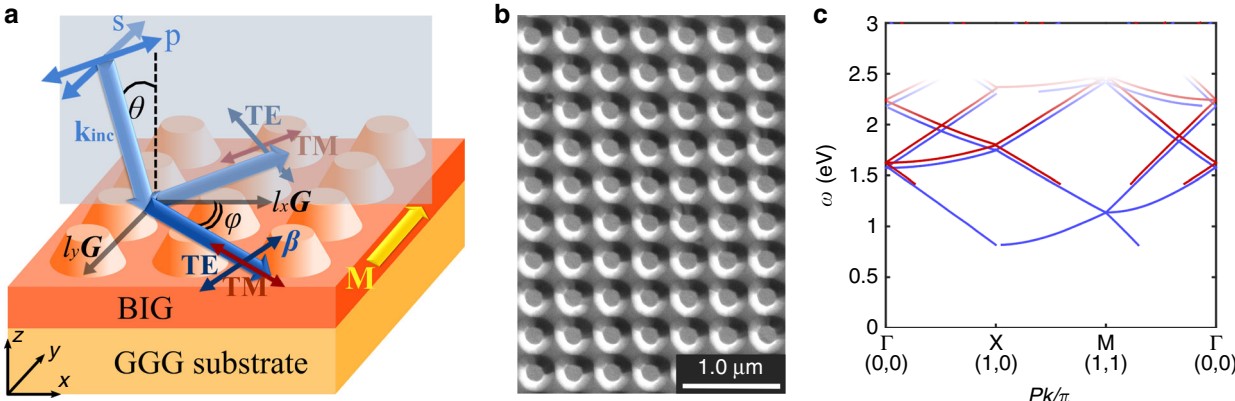

**Fig. 1 Magnetic metasurface. a** Scheme of the sample and mode excitation; **b** SEM image of the fabricated structure. **c** Dispersion diagram for the TM- (red color) and TE- (blue color) modes in 2D magnetic metasurface with a smooth BIG underlayer. Line whitening corresponds to the absorption frequency range of the BIG material.

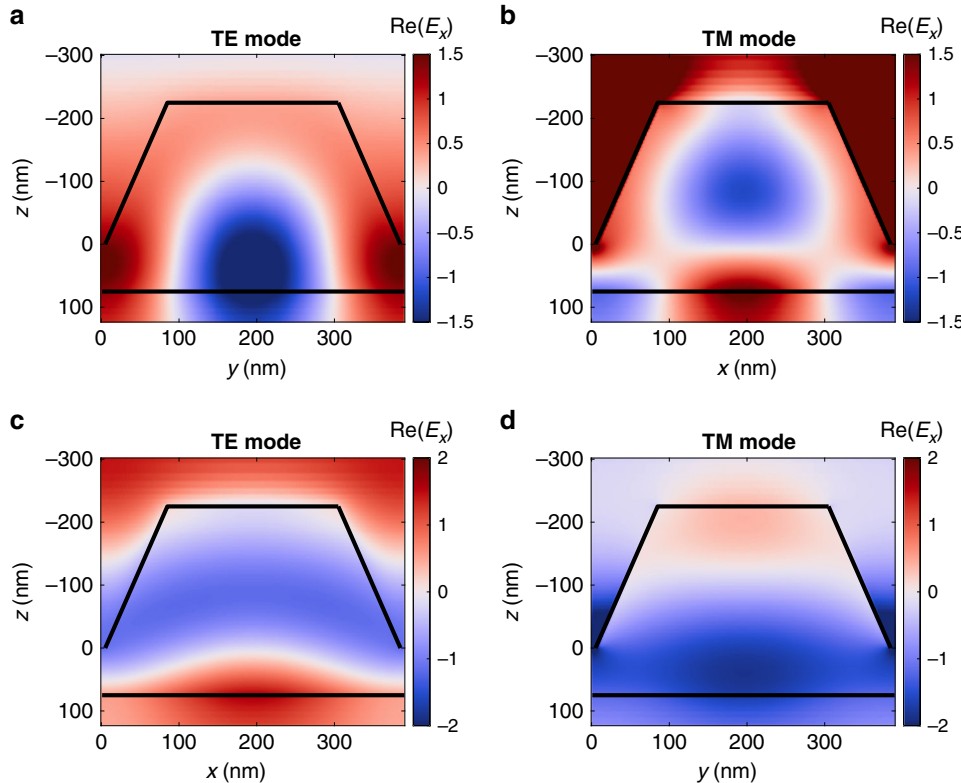

**Fig. 2 Electromagnetic field distribution Re($E_x$) inside a nanopillar of magnetic metasurface. a, c** The TE(0,1)-mode propagating in y-direction and **b, d** the TM(1,0)-mode propagating in x- direction, excited by p-polarized light with $E = (E_x, 0, 0)$ at normal incidence. All the cross-sections are taken at the center of the nanopillar. One period of magnetic metasurface is shown. Images show the field distribution in a normal to sample surface plane: **a, b** images show the field distribution in the direction along the wave vector and **c, d** show the field distribution in a plane orthogonal to the wavevector of the modes.

$D = 200$ nm were arranged to form a square array with the period $P = 390$ nm (Fig. 1a). The nanopillars have a height $H = 225$ nm, and below them, a thin BIG smooth underlayer of thickness $h = 75$ nm was left after patterning. An SEM image of the structure shows that the resulting magnetic metasurface structure is regular, and the sidewalls of nanopillars are slanted at an angle (Fig. 1b).

Such magnetic metasurface structure supports quasi-waveguide mode excitations which propagate along the structures as through a smooth effective dielectric slab and scatter on the nanopillars (see Fig. 2). Treating the structure of 'iron-garnet film + nanopillars + air gaps' as the planar homogeneous guiding layer (with the effective refractive index $n_{wg}$) surrounded by air and the GGG substrate as the semi-infinite claddings, one may apply the planar waveguide theory[48] to it and get the estimation of the propagation constant $\beta$ for the TE or TM guided modes of order $N$:

$$k_0 \sqrt{n_{wg}^2 - \left(\frac{\beta}{k_0}\right)^2}(h+H) - \tan^{-1}\left(\frac{n_{wg}}{n_{air}}\right)^\chi \frac{\sqrt{\left(\frac{\beta}{k_0}\right)^2 - n_{air}^2}}{\sqrt{n_{wg}^2 - \left(\frac{\beta}{k_0}\right)^2}}$$
$$- \tan^{-1}\left(\frac{n_{wg}}{n_{GGG}}\right)^\chi \frac{\sqrt{\left(\frac{\beta}{k_0}\right)^2 - n_{GGG}^2}}{\sqrt{n_{wg}^2 - \left(\frac{\beta}{k_0}\right)^2}} = \pi N \quad (1)$$

where $k_0 = 2\pi/\lambda$ is the free-space wavenumber and $n_j$ is the corresponding refractive index of air ($n_{air}$ or GGG substrate ($n_{GGG}$), $n_{wg}$ is the effective refractive index of the guiding layer, $\chi$ is the coefficient different for TE ($\chi = 0$) and TM ($\chi = 2$) polarizations. This planar waveguide approximation with the estimated value

$n_{wg} = 2.23$ shows a very good agreement with experimental data, as one can see in Fig. 3. According to Eq. (1) due to the small thickness of the BIG film, the structure supports only TM$_0$ and TE$_0$ modes in the transparent region of the BIG ($\lambda > 500$ nm). The cutoff wavelength for the TE$_0$ mode is $\lambda_c^{TE} = 1500$ nm, and for the TM$_0$ mode $\lambda_c^{TM} = 865$ nm.

The nanopillar lattice, on the one hand, is used to achieve coupling between the incident light and the quasi-waveguide mode via its diffraction on the 2D periodic lattice. Due to the 2D periodicity of the magnetic metasurface, the propagation direction of the quasi-waveguide mode could be almost arbitrary in the surface plane depending on the angle of incidence $\theta$ and wavelength of the laser beam, according to the following phase-matching condition:

$$\mathbf{k}_\tau^{inc} + l_x \mathbf{G_x} + l_y \mathbf{G_y} = \beta, \quad (2)$$

where $\mathbf{k}_\tau^{inc} = k_0 \sin\theta \mathbf{e_x}$ is the projection of the wavevector of the incident light on the sample plane, $l_x$ and $l_y$ refer to the evanescent diffraction orders exciting the mode, and $\mathbf{G}_x$ and $\mathbf{G}_y$ are reciprocal lattice vectors along two orthogonal directions of the nanopillar array, $|\mathbf{G}_x| = |\mathbf{G}_y| = 2\pi/P$. The guided mode excited by $l_x$ and $l_y$ diffraction orders are denoted as $(l_x, l_y)$ Fig. 1c.

The electromagnetic field distribution was studied numerically for the TM- and TE- modes excited in the structure under illumination by the p-polarized light, namely, optical vector **E** in x-z plane (Fig. 2). For the TE-mode (Fig. 2a, c) it is clearly seen that the major part of the mode energy is concentrated in the smooth part of the structure due to the high refractive index of the GGG substrate $n_{GGG} = 1.95$[49]. For the TM-mode, the electromagnetic field distribution is a bit more complicated (Fig. 2b, d). The nanopillar lattice itself serves as a medium for the

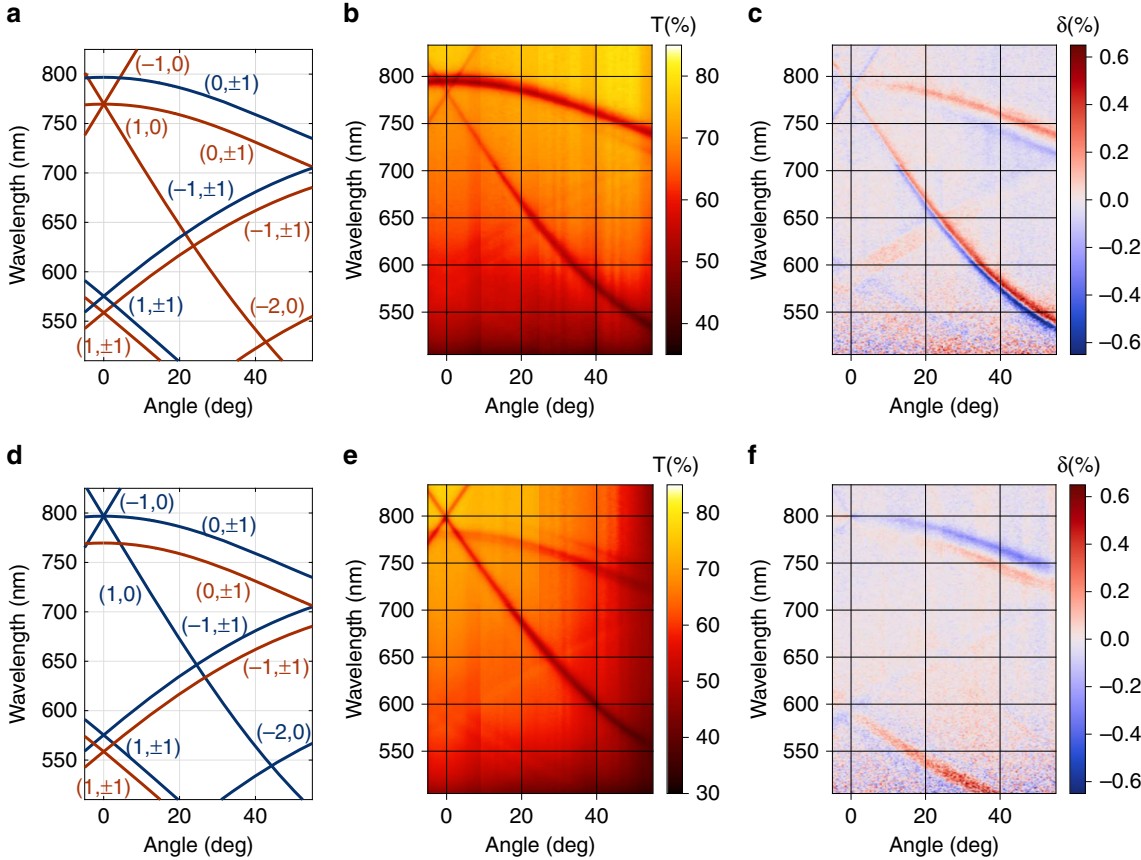

**Fig. 3 Wavelength vs. incidence angle spectra of magnetic metasurface for p- (upper panel) and s- (lower panel) polarizations of the incident light.**
**a**, **d** TM- (red color) and TE- (blue color) mode dispersion in magnetic metasurface calculated according to Eqs. (1) and (2). Experimental **b**, **e** transmittance and **c**, **f** magnetooptical intensity modulation $\delta$ spectra of the magnetic metasurface.

mode propagation, so that the TM-mode field significantly penetrates in the nanopillars. Another consideration, proving that the presence of the nanopillars plays an important role in the mode formation, is that such a thin film with h=75 nm could not support the TM$_0$ modes itself ($\lambda_c^{TM} = 483$ nm for $n_{BiIG} = 2.5$). The penetration of the propagating TM-mode in the nanopillars increases the scattering of the mode, and leads to a decrease in the magnitude of the observed TM-resonances in the transmittance spectra compared to the TE-resonances (see Fig. 3b, e).

The important feature of the 2D magnetic metasurface is that both TM and TE modes propagating at an angle to the incidence plane (with $l_x \neq 0$) could be excited by both p- and s- polarized incident light since p- and s- polarizations contain non-zero projections of electromagnetic field on eigen polarization vectors of the TM- and TE-modes (see scheme in Fig. 1a). For example, at $\Gamma$-point ($\theta = 0$, Fig. 1c) s-polarized light (incident $\mathbf{E}_{inc}$ is along y-axis) could excite not only the TE-mode propagating in the plane of incidence ($l_x = \pm 1, l_y = 0$), but also the TM-mode that propagates perpendicular to the plane of incidence ($l_x = 0, l_y = 1$), or the TM-mode, or TE-mode propagating in diagonal directions ($|l_x| = |l_y| = 1$). All of these modes have different wavelengths according to Eqs. (1) and (2), see also Fig. 3d. Notice also that modes with $+l_y$ and $-l_y$ are always simultaneously excited due to the structure symmetry with respect to the plane of incidence. For p-polarized light the situation is vice versa, see Fig. 3a, b.

Indeed, all of the TM- and TE-modes propagating outside the incidence plane $l_y \neq 0$ are observed in the transmittance spectra for both polarizations (Fig. 3b, e), while the modes propagating in

the incidence plane ($l_y = 0$) are observed only for a corresponding polarization of the incident light: TM-modes – for the p-polarized illumination, and TE-modes for the s-polarized one.

The subwavelength period of the nanopillar array makes the structure resonant: only the zeroth propagating diffraction order exists in transmission and reflection. In this case, the effect of mode excitation is directly seen as dips in the transmittance spectra without being distributed between multiple channels like in the case of the non-resonant gratings with larger period[50]. Such resonant behavior makes the optical spectra very sensitive to the excitation of the propagating modes by evanescent orders. However, similar to the ordinary diffraction patterns observed for propagating diffraction orders, the higher the evanescent diffraction order is, the lower amplitude it possesses. Therefore, guided modes with $|l_x| = 1, l_y = 0$ (or vice versa) are clearly seen in transmittance spectra; modes with $|l_x| = 1, |l_y| = 1$ are seen significantly weaker; while modes with $|l_x| = 2$ totally vanish (Fig. 3b, e).

It should be noted that the observed transmittance dips are very narrow. The Q-factor of the guided-mode (1,0) resonance found from the transmittance is about $Q = \frac{\lambda}{\Delta\lambda} = 110$. It is almost one order of magnitude larger than the one for the magneto-plasmonic structures reported earlier[14,36–38]. At the same time, the transmittance of the structure is also high $T \approx 70\%$, and the resonances are up to $\Delta T \approx 30\%$ magnitude. These values are almost one order higher than for the magnetoplasmonic structures with lossy metallic layers (for example, $T \approx 1\%$ and $\Delta T \approx 0.5\%$ were reported in ref. [38]).

Such narrow resonance width makes the magnetic metasurface structure extremely sensitive to the external magnetic fields, while

high values of the transmittance makes it promising for various practical applications.

**Transverse magnetophotonic intensity effect (TMPIE).** The quasi-guided modes are sensitive to the magnetization of the structure and therefore to the external magnetic field. Even slight modifications of the optical field distribution or the mode dispersion drastically change the far-field response of the structure, in particular, the transmittance and reflectance spectra. Therefore, the modes excitation enhances the magneto-optical effects and opens up a way for efficient control of light propagation through the nanostructured media via magnetic field.

As it will be shown further, in the case of the magnetic metasurface considered here, guided modes are evolved and magnetic-field-induced modification of the guided modes plays a crucial role in the formation of the magneto-optical response. Moreover, some configurations provide the magnetooptical intensity effect totally absent if the guided-mode supporting magnetic metasurface is substituted by a smooth iron-garnet film. We study the phenomenon of the magnetooptical light intensity modulation in transmission, defined as the relative change in transmittance due to the magnetization switching in the perpendicular to the incidence plane direction:

$$\delta = 2\frac{T(+\mathbf{M}) - T(-\mathbf{M})}{T(+\mathbf{M}) + T(-\mathbf{M})} \times 100\%. \tag{3}$$

Note that we focus our attention on the intensity effect in transmission, however a similar effect would be observed in reflectance as well.

Let us first discuss modes propagating in the incidence plane of the light ($\beta_y = 0$). In this case the transversal magnetization **M** nonreciprocally changes the propagation constant of a TM-mode $\beta^{TM} = \beta_0^{TM} + \Delta\beta^{TM}(\mathbf{M})$ and leaves a TE-mode totally unaffected[31]. The related far-field magnetooptical intensity effect is usually referred as TMOKE (transverse magnetooptical Kerr effect) both in transmission and in reflection.

No odd magnetooptical effect is seen at the corresponding (1,0) TE-mode in spectra for s-polarization in spite of a very sharp resonance in transmission (Fig. 3e, f). At the same time, for (1,0) the TM-mode excited in p-polarization the TMOKE response is well-pronounced (Fig. 3c). Its spectrum has an S-shape ($\delta_{min} \approx -\delta_{max}$), confirming that the origin of TMOKE for this mode is a nonreciprocal resonance shift. For the same smooth (unetched) iron-garnet film the measured TMOKE value at 780 nm wavelength is less than $\delta_T = 5 \cdot 10^{-3}\%$ in transmission (and TMOKE in reflection is $\delta_R = 2 \cdot 10^{-2}\%$). Therefore, magnetic metasurface produces about two orders of magnitude increase of TMOKE in transmission compared to the smooth film.

However, Fig. 3c, f clearly show that magnetooptical light modulation is observed not only for TM- but for TE-modes as well with the similar magnitude, therefore such cases should be analyzed in detail.

The situation when the excited mode propagates at an angle $\varphi = \tan^{-1}\left(\frac{l_y}{l_x + k_\tau^{inc}/G_x}\right)$ to the incidence plane (Fig. 4a) is more complicated. As it was mentioned above, guided modes with $+\varphi$ and $-\varphi$ are always excited simultaneously, with amplitudes equal to each other in the absence of the magnetic field; also, both p- and s- polarizations of the incident wave could couple to both TM- and TE- guided modes. It is important that, "transversal" with respect to the incidence plane magnetization **M** really has both transversal, $M_t$, and longitudinal, $M_l$, components with respect to the mode propagation direction: $\mathbf{M} = M_t\mathbf{e}_\perp + M_l\mathbf{e}_{||}$. Therefore, there are two factors that should be taken into account

to analyze the transverse magnetophotonic intensity effect (TMPIE) arising in this configuration.

1. The longitudinal magnetization $M_l = M \sin\varphi$ mixes the $E_x$, $E_z$ and $E_y$ components of the light propagating in the magnetic metasurface. The guided eigenmodes of the iron-garnet layer no longer have pure TM ($E_y = 0$) or TE ($E_x = E_z = 0$) polarizations[51,52] as in the non-magnetized case. Both TM- and TE-polarized modes acquire small TE- or TM-polarized additional components proportional to $M_l$[51,52] without change of the mode propagation constant (in the linear in $M$ approximation). Therefore, these modes should be referred as quasi-TM and quasi-TE, respectively, with the same propagation constants $\beta_{TM}$ and $\beta_{TE}$ as in the non-magnetized case. It is important to highlight that here we deal with just a modification of the eigenmode polarization for each of the modes rather than to their coupling which could be observed only for the relatively thick films with close propagation constants for the two modes: $\beta_{TM} \approx \beta_{TE}$[1].

Due to the emergence of these magnetization-induced components, the tangential component $\mathbf{E}_\tau^{wg} = \{E_x, E_y\}$ of the propagating guided mode is no longer perpendicular (for quasi-TE) or parallel (for quasi-TM) to $\boldsymbol{\beta}$. Small magnetization-induced additional components of the orthogonal polarization tilts $\mathbf{E}_\tau^{wg}$ at an angle $\alpha$ (see Fig. 4a for TE-polarized mode) with respect to its direction in non-magnetic case. As the vectors of the incident light polarization tangential component $\mathbf{E}_\tau^{inc}$ and eigenmode polarization $\mathbf{E}_\tau^{wg}$ are oriented at some angle to each other ($\varphi$ or $\frac{\pi}{2} - \varphi$, namely), this magnetization-induced tilt of $\mathbf{E}_\tau^{wg}$ results in a variation of the angle between these vectors ($\varphi + \alpha$ or $\frac{\pi}{2} - \varphi - \alpha$, namely). Therefore, the efficiency of mode excitation, which is proportional to the projection of $\mathbf{E}_\tau^{inc}$ on $\mathbf{E}_\tau^{wg}$ and thus depend on the angle between them, also changes. For example, as it is clearly seen from Fig. 4a, $-M_l$ would cause p-polarized incident light (which polarization is shown by a blue arrow) couple to TE-mode better than $+M_l$. The magnetization-induced rotation $\alpha$ has an opposite sign for the mode propagating with $+\beta_y$ and $-\beta_y$, and for longitudinal magnetization components $+M_l$ and $-M_l$ (see light and dark blue arrows in Fig. 4a corresponding to $\mathbf{E}_\tau$ electromagnetic field component for $+\mathbf{M}$ and $-\mathbf{M}$ directions). Therefore, the magnetic-field induced variation of the coupling of incident light to the excited mode have the same sign and value for $+\beta_y$ and $-\beta_y$ mode. This results in the magnetic-field induced variation of the coupling efficiency of the incident p- or s- polarized light to quasi-TM or quasi-TE modes, which, in turn, changes the transmittance of magnetic metasurface in an odd way in the magnetization. It is also rather obvious that if the magnetization-induced rotation of the mode polarization results in the increase of the coupling with the p-polarized incident light, it results in the decrease of the coupling with the orthogonal s-polarized incident light polarization as well, and vice versa. Similarly, if the coupling of the incident light to the TM-mode increases due to structure magnetization, it simultaneously decreases the coupling to the TE-mode, and vice versa. This explains the different signs of the TMPIE observed for different modes in Fig. 3c, f.

For example, let us consider the resonance branches that correspond to modes excited by equal diffraction order $(l_x, l_y)$, (0,1) mode in particular. TMPIE values obtained for a p- polarization of incident (Fig. 3c) light for (0,1) TM- and (0,1) TE- mode have opposite signs and similar magnitudes. TMPIE for (0,1) TM- and (0,1) TE- mode excited by

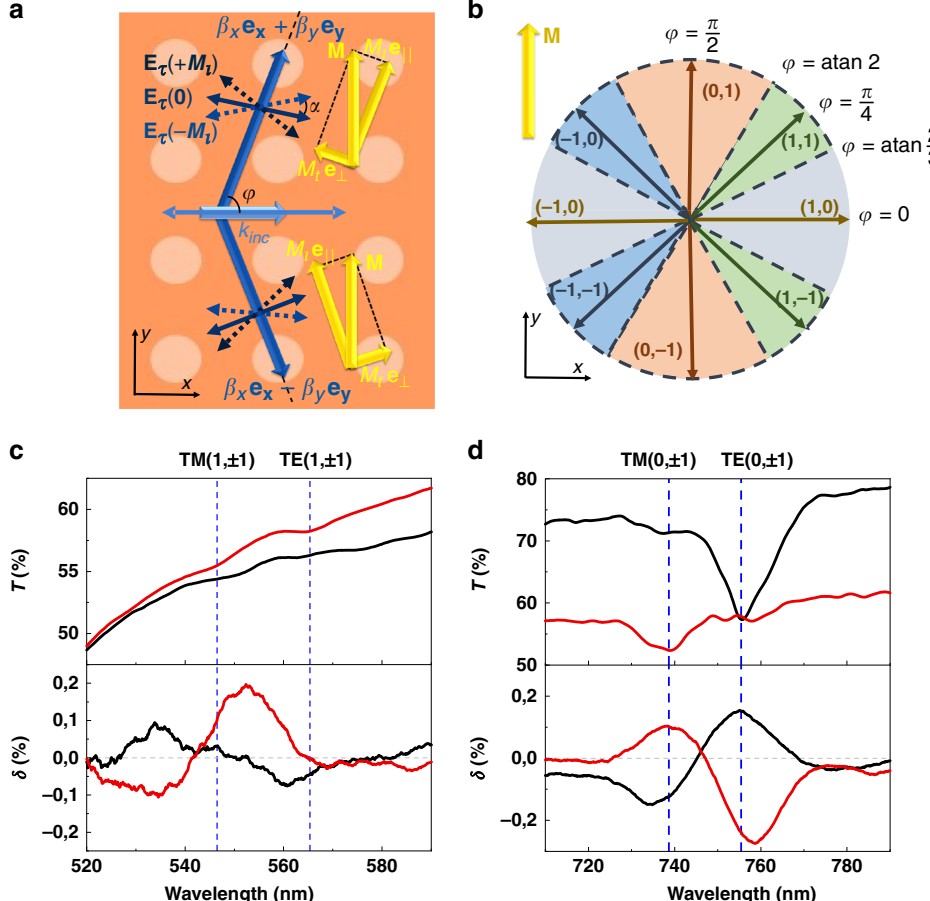

**Fig. 4 Transverse magnetophotonic intensity effect (TMPIE) in magnetic metasurface. a** Schematic representation of the physical mechanism of the effect for the guided modes propagating outside the incidence plane (shown for the TE-modes excited by p-polarized light). **b** Schematic representation of $(l_x, l_y)$ mode propagation direction described by angle $\varphi$, arrows show $\varphi(\theta = 0)$ and sectors of corresponding colors show all possible range of $\varphi$ for the mode. Simultaneously excited modes are shown in similar color. **c, d** Wavelength spectra of TMPIE and transmittance for p- (black color) and s- (red color) polarized incident light shown in different spectral regions at the fixed angles of incidence: 44 deg for **c** and 15 deg for **d**.

incident s- polarization (Fig. 3f) has exactly the same peculiarity. However, the sign of TMPIE for (0,1) TE-mode is opposite for p- and s-polarizations of incident light, and the sign of TMPIE for (0,1) TM-mode as well. This TMPIE spectral feature also confirms the proposed mechanism of magnetooptical effect based on polarization-induced coupling variation.

2. The impact of the transversal magnetization component $M_t = M \cos\varphi$ also should be taken into account. It is almost negligible for (0,1) modes because their propagation direction corresponds to very small $M_t$ values (see scheme in Fig. 4b). However, for $(+1, +1)$ modes $M_t$ is significantly higher. Due to the polarization transformation inside the magnetic metasurface, both polarizations acquire TM-components so that $M_t$ also makes small non-reciprocal shift of the guided mode resonance for the guided quasi-TM- and quasi-TE- modes. Although this mechanism is obviously more complicated than for the pure TMOKE discussed above for the in-plane modes, experimental TMPIE spectra (Fig. 3c, f) show a good agreement with this qualitative theory. Indeed, TMPIE resonances for all (0,1) modes are U-shaped, as it is expected from the magnetic-field induced modulation of the resonance depth.

On the contrary, TMPIE resonances for $(l_x \neq 0, l_y)$ experience $M_t$-induced shifts and are S-shaped. This effect is clearly observed in Fig. 3f: for the $(-1, \pm1)$ TM and TE modes, the S-shaped

TMPIE responses of opposite signs almost diminish in the region between the two resonances. For the $(+1, \pm1)$ modes, the situation is opposite: S-shaped TMPIE responses have the same sign in the region between the two modes, and this results in the significant enhancement of the TMPIE response.

## Discussion
The magneto-optical response of the nanopatterned iron-garnets was experimentally studied and revealed a major potential of all-dielectric magnetophotonic metasurfaces. It is demonstrated that magnetic metasurfaces based on dielectric materials could exhibit novel phenomena fundamentally impossible in other structures.

In particular, we have experimentally found a novel linear magnetooptical intensity effect, TMPIE, in a magnetic metasurface structure consisting of iron-garnet nanopillars on a thin iron-garnet film. The intensity modulation of the transmitted light in the magnetic metasurface magnetized transversally with respect to the incidence plane of the light is demonstrated. Notably, this effect is completely absent for the smooth iron-garnet film. Moreover, other nanostructures implemented up to this date, allow controlling of only p-polarization of the incident light. Thus, the unique feature of the observed TMPIE is that in contrast to other magneto-optical intensity effects, it arises for both p- and s- polarizations of the incident light with a similar magnitude, and is significantly enhanced by excitation of both TM- and TE-polarized modes of the magnetic metasurface.

TMPIE in 2D magnetic metasurface is important for various practical applications, where it is necessary to perform magneto-optical measurements in both optical polarizations simultaneously. For example, measurement of the magneto-optical response in the two polarizations in biosensing nanostructures might significantly enhance performance of the sensor and also distinguish bulk and surface effects and therefore could treat properly the variations of the bulk refractive index such as due to the temperature fluctuations. Vector magnetometers also could benefit greatly from the possibility to observe the magneto-optical signal related to the two orthogonal polarizations sensitive to the in-plane magnetic field components. Magnetooptical intensity effect observed for both polarizations of the excited modes is also important for the integrated optics devices. Modulation of the propagating guided modes, especially the TE-modes that are the fundamental modes of the waveguides, are essential for data processing applications. These are just few examples of the novel opportunities opened up by the proposed magnetic metasurface in various nanophotonic devices.

## Methods

**Sample fabrication**. A wafer of epitaxially grown two-µm-thick film of BIG ($Bi_{0.7}Gd_{0.3}Lu_{2.0}Ga_{0.8}Fe_{4.2}O_{12}$) on gadolinium gallium garnet (GGG) was cut into 1 cm × 1 cm pieces via dicing saw. The film was grown by liquid-phase-epitaxy on (001)-oriented GGG substrate at II-VI, Inc. Samples were cleaned by sonication in acetone/IPA/DI water spray and blow-dried with nitrogen. The BIG film thickness was reduced from 2 µm to 300 nm by wet etching in ortho-phosphoric acid (85% from Sigma-Aldrich) for about an hour. The acid was heated to 130 °C and stirred using a magnetic stirrer for uniform etching of the film. The thickness of the film was measured using V-vase ellipsometry. Nano-patterns were then fabricated using a 100 KeV e-beam lithography system (VISTEC EBPG 5000+). A 250 nm-thick ZEP positive e-beam resist was spin-coated on the substrate together with a 30 nm-thick Au layer on top to suppress electrical charging of the dielectric garnet film during electron-beam exposure. 2D circular disk patterns were patterned on ZEP resist by uniform e-beam exposure at a dose of 150 µC/cm$^2$ under proximity effect correction (PEC). The Au layer was first removed by wet etching in a gold etchant solution and afterwards the resist was developed in an amyl acetate solution. The resist patterns were then transferred onto the BIG film by sputter-etching in an argon-ion milling system (Intlvac Nanoquest) with ion source parameters: beam voltage – 200 V, accelerating voltage – 24 V, beam current – 70 mA and plasma forward power – 72 Watt. The sample stage temperature was kept at 6 °C throughout the etching duration of about 90 min. The resist was then removed using resist remover N-methyl-2-pyrrolidine (NMP) by heating at 80 °C for about half an hour.

Finally, the sample was wet-etched in phosphoric acid at 130 °C for ~1 min to smoothen the side walls of the nanopillars. The higher etch rate towards the tip of the nanopillars due to stirred acid, produced the truncated-cone shape of the nanopillars. The roughness of the wet-etched surface is usually less than 1 nm for wet-etched film, however the trade-off here is the imperfect nanopillars. The AFM images are shown in Supplementary Note 3.

The film has in-plane anisotropy with in-plane saturation field of $H_{in}$ = 30 Oe, out-of-plane saturation field of $H_{out}$ = 2 kOe, and saturation magnetization $4\pi M_s$ = 860 G.

Iron-garnet film was described by the following spectral dependence of permittivity and gyration constant:

$$\varepsilon(\lambda) = (0.1207 \cdot \lambda + 0.9119) \cdot \left( 1.07 + \frac{4.90}{1 - \left(\frac{0.303}{\lambda}\right)^2} + \frac{0.12}{1 - \left(\frac{0.494}{\lambda}\right)^2} - \frac{0.543}{\lambda} \right),$$

(4)

and $g = 0.8 \cdot 10^{-2} - 0.004\lambda + 6.4 \cdot 10^{-6}\lambda^2 - 5 \cdot 10^{-9}\lambda^3 + 14 \cdot 10^{-13}\lambda^4$, ($\lambda$ in µm) which was obtained previously for a similar smooth film[38].

**Experimental setup**. Transmittance and TMPIE spectra were measured using the following experimental setup. A tungsten halogen lamp was used as a light source. Light passed through fiber with exit diameter of 200 µm to obtain a homogeneous point-like source. Then it was collimated using an achromatic 35-mm lens, polarized using a Glan-Taylor prism, and focused onto the sample with a 20x objective. The light passed through the sample, and then it was collimated with a 20x objective, and then focused to the spectrometer slit with a pair of 15- and 30-mm lenses, so each point of slit corresponded to a certain angle of light incidence angle onto the sample. A spectrally and angularly decomposed light was detected using a 2D charge-coupled device. For magneto-optical measurements the sample was placed in a magnet with a maximum magnetic field of 3 kOe.

**Numerical simulations**. Electromagnetic simulation of modes propagation inside the unit cell of the considered all-dielectric structure was carried out by numerical solution of Maxwell equations using the rigorous coupled-wave analysis (RCWA) approach[53,54].

For simulations the dispersion of dielectric permittivity and gyration of BIG ($\varepsilon_{BiIG}$ and $g$ correspondingly) was taken into account, so that $\varepsilon_{BIG} = 7.3057 + 0.034i$ and $g_{BIG} = 0.47 \cdot 10^{-2} + 0.0005i$ for the wavelength of 700 nm. The refractive index of GGG was equal to 1.97.

## Data availability

The data that support the plots within this paper and other findings of this study are available from the corresponding author Daria Ignatyeva (ignatyeva@physics.msu.ru) upon reasonable request.

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

## Acknowledgements

This work was financially supported by the Russian Ministry of Education and Science, Megagrant project N 075-15-2019-1934. The device fabrication was conducted at the Nano-fabrication facility, Nanocenter, University of Minnesota and the microfabrication facility (MFF) at Michigan Tech. University. M.L. and D.K. gratefully acknowledge support from the Michigan Tech Henes Center for Quantum Phenomena. The authors acknowledge Artem K. Grebenko from LNM, CPQM, SkolTech for AFM studies. The authors thank Dr. N.E. Khokhlov for the fruitful discussion.

## Author contributions

D.O.I and V.I.B. conceived and designed the experiments. D.K. fabricated the magnetic metasurface and performed SEM and thin film metrology. A.A.V. and M.A.K. measured the wavelength- and angle-resolved spectra of the transmittance and magneto-optical intensity modulation. A.I.C. performed the AFM study of the metasurface. D.O.I. performed the analytical description. D.M.K. performed the numerical analysis. All of the authors contributed to the discussion of the data. D.O.I. wrote the manuscript with contributions from V.I.B., D.K. and M.L. The work was coordinated by V.I.B.

## Competing interests

The authors declare no competing interests.
