## [Peer Review File · Nature Communications]

REVIEWER COMMENTS

Reviewer #1 (Remarks to the Author):

This article deals with the 2D nano-structuration of magneto-optical (MO) yttrium iron garnet based material. The main result concerns the experimental evidence of transverse magnetic Kerr effect for both p and s polarization and for this it deserves to be published. The 0.25% maximum of TMOKE effect is much higher than classical TMOKE effect on smooth surface, but less than maximum TMOKE effect observed in literature (around ~10%, but only for TM polarization).

However, this is subjected to clarifying or correcting the following remarks :

1-TMPIE is defined page 9, but it is used before (page 6)

2-Authors write page 2 : "However, fabrication of such complex shapes is challenging, and, on the other hand, the

resulting optical response would be significantly affected by material losses, surface roughness and fabrication inaccuracies neglected in these theoretical studies [37-40]." First, it seems that in ref 37, the material losses are taking into account, and the authors could check this point in the experimental results of these works published recently.

Second, to my mind, some morphological characterizations are missing (AFM for surface roughness or at least tilted SEM). Indeed, on the SEM image (figure 1), it appears that nanopillars look like percolated truncated cones, and not only slightly conical.

3- in equation 1, how is calculated the 2.23 effective refractive index? this value may be affected by the non cylindrical shape of nanopillars

4- In the presented results, TPMIE is expressed in % but not in equation 3

5- some axis titles are missing

6- I would appreciate a superimposed transmittance spectra on figure 4, in order to see effectively the quality factor Q

7- the authors write page 3: "The film can be fully magnetized inplane by magnetic field of $H_{in}=30$ Oe." and page 12 "Out-of-plane saturation field of $H_{out}=2$ KOe" please clarify

8- Some indications on spectral, or at least the lower and upper values, material properties (real and imaginary part): refractive index and MO, would be appreciated.

Reviewer #2 (Remarks to the Author):

The authors report an all dielectric magnetic metasurface based on BIG thin films grown on GGG substrates. High Q resonance is observed for grating coupled TE and TM modes in this device. They observe 2 orders of magnitude higher TMOKE effect compared to a bare BIG thin film. In particular, they discovered a novel MO effect, TMPIE caused by magnetic field tuning of coupling efficiency between free space light and TE and TM modes, which appears to be essentially due to the Faraday effect. I found these results interesting, and important for developing advanced magneto-nanophotonic devices for more versatility of light control. I have several comments for the authors to respond.

1. In figure 1, the etched BIG thin film appeared to be slanted. Please provide more details about the etch process and why the etch is not perfect. What are the exact size and angle of the slope? Please provide more details such as SEM cross-sectional images of the device. For Fig. 1c, was the imaginary part of the BIG material considered during simulation of the band diagram?
2. In figure 2, the modal simulation part did not consider the slanted sidewall. Can the authors comment how much would the results change by considering this factor? Would the simulated spectrum in Fig. 3 change much due to this effect?
3. Please define all the terms in equation (1). Please also explain how this equation was derived. For the effective index $n_{wg}=2.23$, which mode does this effective index belong to?
4. The enhanced TMOKE is very interesting. However this could be due to both an optical enhancement (low transmittance at certain wavelength) and an MO enhancement (higher field in the MO material). Please comment on the contribution from these two effects.
5. Can the authors comment on the Q factor of the device. What are the contributions of scattering and absorption loss to the Q factor. Would the slanted side wall cause scattering loss? How about the difference between TE and TM modes?
6. The TMPIE is interesting. The authors qualitatively commented its origin. I would strongly recommend making the explanation quantitative. What is the expected angle α from this material and structure, and how does this match with experiments? Also, from numerical simulations, more work can be done by simulating the transmittance spectrum and magneto-optical effect. This could be solved by considering the tensor permittivity of BIG in finite element analysis or FDTD simulations. So that a quantitative compare with experiments can be observed.
7. In the TMPIE section, for the modal coupling between TE and TM due to MI, would this effect compete with the birefringence of these two modes? Would the polarization show a beating pattern along the propagation direction? How does this effect relate to the free space coupling?
8. There are some typos, please consider revise.

Reviewer #3 (Remarks to the Author):

This is a very good manuscript reporting timely results. I totally agree with the claim in the manuscript that 'it might be advantageous to modulate both orthogonal linear polarizations, i.e. p- and s-polarizations of light'. The authors are also right to claim that this might have important practical applications. Overall, this manuscript is a significant contribution to the rapidly developing field of all-dielectric magneto-photonics that goes hand-in-hand with a large and growing body of research on all-dielectric nanophotonics and perhaps on metamaterials and metasurfaces. Although I do not always agree with the theoretical interpretation by the authors, they published many works explaining their point of view in the past and this appears to be well-accepted by the community.

Therefore, I would recommend this present manuscript for publication with minor suggestions (see below).

1) I am not really surprised to see that nanostructure-assisted magneto-optical effects in transmission can be observed not only for p-polarisation but also for s-polarization of the incoming light. For instance, for simplicity let us consider the transmission of light through subwavelength apertures in metal films. (Similar effects also occur in dielectric nanostructures under appropriate conditions.) Let us also focus on the relevant phenomenon of extraordinary optical transmission in periodic arrays. It is known that surface electromagnetic modes play a key role in the emergence of the resonant transmission. See, e.g., F. J. Garcia-Vidal, L. Martin-Moreno, T. W. Ebbesen, and L. Kuipers, *Rev. Mod. Phys.* 82, 729 (2010). Usually, such surface electromagnetic modes are excited using p-polarised light; however, they can also be excited using s-polarised light. The review paper above cites several relevant works. Can the authors comment on this and extend their discussion if appropriate?

2) The term 'metasurface' is very popular. However, too many authors abuse it, which I suspect might be also the case of this work. Please justify why the proposed structure is a metasurface.

3) Regarding the 'Transverse magnetophotonic intensity effect'. Can the authors justify the need to call this a new effect? As I said above, this seems to be the same effect but for another light polarisation.

Sincerely,

...

First of all, we would like to thank all referees for their positive evaluation of our work and comprehensive analysis of the manuscript, resulting in important and stimulating comments which greatly helped us in making the presentation stronger. We agree with most of the suggestions and revised the manuscript accordingly. Changes are highlighted in green color.

Our detailed response is as follows:

REVIEWER COMMENTS

Reviewer #1 (Remarks to the Author):

This article deals with the 2D nano-structuration of magneto-optical (MO) yttrium iron garnet based material. The main result concerns the experimental evidence of transverse magnetic Kerr effect for both p and s polarization and for this it deserves to be published. The 0.25% maximum of TMOKE effect is much higher than classical TMOKE effect on smooth surface, but less than maximum TMOKE effect observed in literature (around ~10%, but only for TM polarization).

We really appreciate the supportive conclusion on our work given by the Reviewer-1 and would like to thank him/her for highlighting a main result of the manuscript.

However, this is subjected to clarifying or correcting the following remarks:

1-TMPIE is defined page 9, but it is used before (page 6)
Thank you for the remark. 'TMPIE' on page 6 was changed to "magneto-optical intensity modulation δ " (see page 6).

2-Authors write page 2 : "However, fabrication of such complex shapes is challenging, and, on the other hand, the resulting optical response would be significantly affected by material losses, surface roughness and fabrication inaccuracies neglected in these theoretical studies [37-40]." First, it seems that in ref 37, the material losses are taking into account, and the authors could check this point in the experimental results of these works published recently.

We agree that this sentence is rather confusing in the present form and changed it to "would be significantly affected not only by the material losses, but also by the surface roughness and fabrication inaccuracies that are neglected in these theoretical studies [37-40] ([40-43] in the revised manuscript)" on page 2. Indeed, the resonances in all-dielectric structures are very sensitive to the structure parameters, so that inevitable fabrication inaccuracies significantly deteriorate the resonance widths, Q-factors, and, consequently, TMOKE response. Recent experimental work [44] on magneto-optics of

1D gratings also shows that polarization effects are well enhanced in spite of these inaccuracies, while the TMOKE increase is much less. We have added also references on recent experimental works [44] in the manuscript:

“Although recent experimental studies [44] reveal the enhancement of TMOKE in 1D all-dielectric gratings, it is still observed only for p-polarized illumination and completely vanishes for s-polarized light.” (page 2)

Second, to my mind, some morphological characterizations are missing (AFM for surface roughness or at least tilted SEM). Indeed, on the SEM image (figure 1), it appears that nanopillars look like percolated truncated cones, and not only slightly conical.

The e-beam lithography followed by sputter-etching with argon ions were used for fabricating the structures. To reduce the sidewall roughness after argon-ions sputtering, the sample was wet-etched in heated phosphoric acid at 130°C to smoothen the surface. During wet etching, the acid was stirred using a magnet-stirrer. The higher speed-gradient along the tip of the nanopillars resulted into higher etch rate, thereby forming the shape of the nanopillars as truncated cones, not percolated though.

The SEM image was taken at low accelerating voltage without coating any conducting layer on this insulating garnet material. Small imperfections in the shape has also to do with small charging effect during e-beam exposure/scan. A magnified image below the structure may help to make it more apparent. However, one should bear in mind that there are still some artifacts arising from the SEM representation rather than structural peculiarities.

According to the Reviewer recommendation, we have performed AFM measurements that demonstrate the real profiles of the nanopillars. Atomic Force Microscopy (AFM) imaging was performed with a Bruker Multimode V8 device operating in the Peak Force-HR tapping regime with ScanAsyst HR probes. The tip side angle is 17.5 deg, tip radius is 2 nm.

The estimated vertical angle, based on the average measurements of 128 profiles, is 21.3 ±0.2 deg. AFM shows that nanopillars are not percolated. However, let us notice that below the nanopillars there is a smooth sublayer of the same material so that the presence/absence of the percolation does not significantly affect the structure.

We have added more information about etching impact on the shape of the nanopillars in Methods and provided a Supplementary S1 with these AFM images.

. 3- in equation 1, how is calculated the 2.23 effective refractive index? this value may be affected by the non cylindrical shape of nanopillars

We quite agree that the value of the effective refractive index is sensitive to the non-cylindrical shape of the nanopillars. Indeed, many factors, such as shape of the

nanopillars, dispersion of iron-garnet permittivity itself, etc. should affect this value if calculated carefully. However, this value here is a rough estimation made by the comparison of the experimentally obtained resonance positions shown in Fig.3b,e and numerically calculated using the planar waveguide theory Eqs.(1-2). We aim to provide a ***simple tool for analysis of the spectra of 2D gratings based on the effective value that is the same for all of the modes.*** This approach helps us to ***systemize the observed resonances of the guided modes with different orders (lx,ly) and different polarizations*** rather than to construct some rigorous effective medium theory. Please notice that some effective medium approximations, such as Maxwell-Garnet or Bruggeman, are not strictly applicable to the structure under consideration, as the relative amount of air is about 30%, and the spatial size of the nanopillars is smaller but comparable with the wavelength.

To avoid some misunderstanding, we have clarified this statement:

“Treating the structure of ‘iron-garnet film + nanopillars + air gaps’ as the planar homogeneous guiding layer (with the effective refractive index n_{wg}) surrounded by air and the GGG substrate as the semi-infinite claddings, one may apply the planar waveguide theory [48] to it and get the estimation of the propagation constant β for the TE or TM guided modes of order N .” (pages 3 and 4) and

“This planar waveguide approximation with estimated value of $n_{wg}=2.23$ shows a very good agreement with experimental data, as one can see in Fig. 3.” (page 4)

4- In the presented results, TPMIE is expressed in % but not in equation 3

We have corrected Eq.(3) according to this comment.

5- some axis titles are missing

We have added titles to Fig.3 plots according to this comment.

6- I would appreciate a superimposed transmittance spectra on figure 4, in order to see effectively the quality factor Q

Thank you for this comment. We agree that as the high Q-factor is one of the advantages of the considered metasurface, it is important to show the transmittance spectra for a fixed angle and to study it in details.

We have added Supplementary S3 with the detailed study of the Q-factor, the impact of different kinds of losses and light parameters on it.

7- the authors write page 3: "The film can be fully magnetized inplane by magnetic

field of $H_{in}=30$ Oe." and page 12 "Out-of-plane saturation field of $H_{out}=2$ KOe" please clarify

We have clarified the sentence in Methods section:

"The film has in-plane anisotropy with in-plane saturation field of $H_{in}=30$ Oe, out-of-plane saturation field of $H_{out}=2$ KOe, and saturation magnetization $4\pi M_s=860$ G."

These values mean that one can use a very low magnetic field of 30 Oe to manipulate the magnetization in the sample plane (for example, we switched magnetization between $+M_y$ and $-M_y$ directions that correspond to the transversal configuration). The field $H_{out}=2$ KOe characterizes the external field that should be applied to magnetize the sample perpendicular to its surface in the Faraday configuration. Such Faraday configuration was not used in the present manuscript but is an important characteristic of the iron-garnet film so this information was added to Methods section.

8- Some indications on spectral, or at least the lower and upper values, material properties(real and imaginary part): refractive index and MO, would be appreciated.

We have added the information on the ε and g values in Methods:

Iron-garnet film was described by the following spectral dependence:

$$\varepsilon(\lambda) = (0.1207 \cdot \lambda + 0.9119) \cdot \left(1.07 + \frac{4.90}{1 - \left(\frac{0.303}{\lambda}\right)^2} + \frac{0.12}{1 - \left(\frac{0.494}{\lambda}\right)^2} - \frac{0.543}{\lambda} \right),$$

and $g = 0.8 - 0.004\lambda + 6.4 \cdot 10^{-6}\lambda^2 - 5 \cdot 10^{-9}\lambda^3 + 14 \cdot 10^{-13}\lambda^4$, (λ in μm) which was obtained previously for a similar smooth film [38].

Reviewer #2 (Remarks to the Author):

The authors report an all dielectric magnetic metasurface based on BIG thin films grown on GGG substrates. High Q resonance is observed for grating coupled TE and TM modes in this device. They observe 2 orders of magnitude higher TMOKE effect compared to a bare BIG thin film. In particular, they discovered a novel MO effect, TMPIE caused by magnetic field tuning of coupling efficiency between free space light and TE and TM modes, which appears to be essentially due to the Faraday effect. I found these results interesting, and important for developing advanced magneto-nanophotonic devices for more versatility of light control. I have several comments for the authors to respond.

We would like to thank the Reviewer-2 for positive evaluation of our results and for noticing their promise for advanced nanophotonics.

1. In figure 1, the etched BIG thin film appeared to be slanted. Please provide more details about the etch process and why the etch is not perfect.

The nanostructures were patterned by e-beam lithography and afterwards sputter-etching with accelerated argon-ions towards the sample surface (also described in detail in the manuscript under Methods “sample fabrication”). The sputter-etching results into rough sidewall surface of the nanostructures. Therefore, the sample was wet-etched in phosphoric acid heated at 130°C to reduce the roughness. During this wet etching- step, the acid was stirred using a magnet-stirrer. This sets up the velocity-gradient and thus the etch-rate-gradient normal to the sample surface as the sample was held normal to whirling axis of the acid. The higher etch rate towards the tip of the nanopillars produced the truncated-cone shape of the nanopillars. The roughness of the wet-etched surface is usually less than 1 nm for wet-etched film, however the trade-off here is the imperfect nanopillars.

We have added more information about etching impact on the nanopillar shape in Methods.

What are the exact size and angle of the slope? Please provide more details such as SEM cross-sectional images of the device.

The angle of the slope has been found as $\sim 21^\circ$ to the film normal on the measurement of the diameters/radii of the truncated-cone structure from SEM images and the 225 nm height of the nanopillars.

According to the Reviewer recommendation, we have performed AFM measurements of cross-section profiles of nanopillars. Atomic Force Microscopy (AFM) imaging was performed with a Bruker Multimode V8 device operating in the Peak Force-HR tapping regime with ScanAsyst HR probes. The tip side angle is 17.5 deg, tip radius is 2 nm. The estimated vertical angle, based on the average measurements of 128 profiles, is 21.5 \pm 0.2 deg

Supplementary S1 containing AFM images was added to show the AFM profiles.

For Fig. 1c, was the imaginary part of the BIG material considered during simulation of the band diagram?

For Fig. 1c the imaginary part of the permittivity of the guiding layer was neglected. Therefore, Fig.1c shows the position of the dispersion curves for different modes and not their width. Fig.1c allows us to determine the directions at which modes propagate at

certain frequency and this information is very useful for further analysis of the mutual orientation of the mode wave vector and the external magnetic field.

We have performed the whitening of the lines just to depict schematically the absorption frequency range of the BIG material on this diagram.

2. In figure 2, the modal simulation part did not consider the slanted sidewall. Can the authors comment how much would the results change by considering this factor?

We have performed simulations of the 2D gratings made of truncated cones instead of cylinders. Due to guided character of the excited waves the exact shape does not affect the electromagnetic field distribution much.

We have replaced images in Fig.2 with the ones corresponding to the truncated cones.

Supplementary S2 was added with a comparison of the electromagnetic fields and spectra of the nanopillars having a cylindrical and truncated cone shape.

Electromagnetic field distribution $\text{Re}(E_x)$ for the TM- and TE-modes in the metasurface with truncated cone (left panel) and cylinder (right panel) shape of the nanopillars. The TE(0,1)-mode propagating in y-direction and the TM(1,0)-mode propagating in x-direction, excited by p-polarized light with $E=(E_x,0,0)$ at normal incidence are shown. All the cross-sections are taken at the center of the nanopillar. One period of magnetic metasurface is shown. Images show the field distribution in a normal to sample surface plane: upper panel images show the field distribution in the direction along the wave vector and bottom panel show the field distribution in a plane orthogonal to the wavevector of the modes.

Would the simulated spectrum in Fig. 3 change much due to this effect?

Fig.3a,d shows the spectral positions calculated through Eqs.(1),(2) using the phenomenological effective refractive index (see the details in the answer to Q3). The variation of the nanopillar shape itself leads to rather moderate variations of the spectral position and depth of the observed resonances (see figure below), since both of the structures support guided modes.

We have added Supplementary S2 to discuss the impact of cylinder vs. truncated cone shape of the nanopillars.

:

The numerical transmittance spectra of the 2D structure with **truncated cones** vs. **cylinders**.

3. Please define all the terms in equation (1).

We have added more details on terms in Eq.(1) on page 4.

Please also explain how this equation was derived.

This equation is derived for a waveguide modes in the 3-layered system consisting of a core with higher refractive index and two semi-infinite claddings surrounding it. Generally, it states, that waveguide modes exist at a certain frequency if the total phase incursion acquired by the light in the direction perpendicular to the interfaces is $2\pi N$. This total phase incursion is a sum of the phase incursion proportional to $2k_z H_{wg}$ (the 1st term in Eq.(1)) and the phase incursion provided by Fresnel reflection from the core/cladding interfaces = $2\text{atan}...$ (the 2nd and 3rd terms in Eq.1).

We have rearranged the terms in Eq.(1) to make their physical meaning more obvious, added a reference on page 4 to one of the books where this equation which is common

for waveguides is derived and explained in more details (Eq.(2.178 in [48]). We have also provided the following additional explanation in the manuscript:

“Treating the structure of ‘iron-garnet film + nanopillars + air gaps’ as the planar homogeneous guiding layer (with the effective refractive index n_{wg}) surrounded by air and the GGG substrate as the semi-infinite claddings, one may apply the planar waveguide theory [48] to it and get the estimation of the propagation constant β for the TE or TM guided modes of order N .” (pages 3 and 4)

For the effective index $n_{wg}=2.23$, which mode does this effective index belong to?

The effective refractive index is used to describe the structure of ‘iron garnet film + nanopillars + air gaps’ system as the planar smooth layer of the same width. It was assumed to be the same for all of the modes. Such approach gave us **a simple tool to systemize the observed resonances of the guided modes with different orders (lx,ly) and different polarizations.** We do not aim to construct some rigorous effective medium theory, while the well-known effective medium approximations, such as Maxwell-Garnet or Bruggeman, are not quite applicable to the structure under consideration, as the relative amount of air is about 30%, and the spatial size of the nanopillars is comparable with the wavelength.

To avoid some misunderstanding, we have clarified this statement:

“This planar waveguide approximation with estimated value of $n_{wg}=2.23$ shows a very good agreement with experimental data, as one can see in Fig. 3.” (page 4)

Please also notice that it is not equal to the refractive index of the modes $n_{\beta} = \frac{\beta}{k_0}$ which is indeed different for TM and TE modes and has spectral dependence as shown below. Notice that this simple model gives results that are in rather good agreement with the experimental results $n_{\beta} = \frac{\lambda_0}{mP}$ providing $n_{\beta} = 2.04$ for TE-mode at $\lambda_0 = 797$ nm (model gives $n_{\beta} = 2.05$) and $n_{\beta} = 1.99$ for TM-mode at $\lambda_0 = 778$ nm (model gives $n_{\beta} = 1.975$).

4. The enhanced TMOKE is very interesting. However this could be due to both an optical enhancement (low transmittance at certain wavelength) and an MO enhancement (higher field in the MO material). Please comment on the contribution from these two effects.

This is an important point, indeed. We agree that as TMOKE(TMPIE) is determined as relative intensity modulation it is important to show that the observed 'enhancement' is not an artifact of low transmittance.

To justify our statement that both TMOKE and TMPIE enhancements arise mainly due to the MO we plotted $\Delta T = [T(+\mathbf{M}) - T(-\mathbf{M})] \times 100\%$ spectra shown below. One may see that these figures just repeat Fig.4c,f.

Actually, the observed transmittance variation due to the guided mode excitation does not exceed 20% (Fig.4b,e) at $T \approx 70\%$ background. Thus, considering in Eq.(3) $\Delta T = \text{const}$, the resonant variation of the transmittance itself gives only $\frac{\delta_{res}}{\delta_{nonres}} = 1.4$ ratio while the observed resonant enhancement of δ is higher than an order of magnitude (see Fig.4c,f). Thus, this optical contribution can be considered negligible in our case.

5. Can the authors comment on the Q factor of the device. What are the contributions of scattering and absorption loss to the Q factor. Would the slanted side wall cause scattering loss?

We agree that it is really an important question and have added a Supplementary S3 with the detailed study of the Q-factor.

Actually, there are 3 main channels of losses for a guided mode excited in the structure: 1) absorption in BIG material 2) leakage of radiation due to the back-coupling to the propagating reflected/transmitted waves 3) scattering losses due to the fabrication inaccuracies. As surface roughness is <1nm and the non-resonant transmittance in the transparency region is close to the one without nanostructure (reflection at the GGG substrate/air interface gives T=80% as the theoretical limit) one can neglect (3) loss channel and assume $Q^{-1} = Q_{abs}^{-1} + Q_{leak}^{-1}$.

Q_{abs} can be roughly estimated as $Q_{abs} = n'_{BIG}/n''_{BIG}$ that gives $Q_{abs} = 954$ at 800 nm (we have added the model of smooth BIG permittivity in Methods section which gives $n_{BIG} = 2.5650 + 0.0028i$). Thus, for experimentally observed Q=109 (TE(1,0)-mode) one may get an estimation $Q_{leak} = 123$. Thus, the main losses and Q-factor limitation comes from the back-coupling of the guided mode to the propagating radiation. This fact was confirmed by the numerical simulations: the resonance widths do not change much if all of the layers are treated as lossless.

The slanted walls affect the Q-factor, indeed. They make the waveguiding layer (BIG layer+nanopillaers) smoother for radiation propagating in the film plane, thus Q-factor increases. See the comparison of the transmittance spectra for the truncated cone and cylindrical shape of nanopillars in the figure below.

Black – with $\text{imag}(\epsilon_{BIG})$

Red – without $\text{imag}(\epsilon_{BIG})$

How about the difference between TE and TM modes?

Indeed, there is a difference in Q-factor for TE and TM modes. One can notice that the TE-resonances are more pronounced in Fig.3b,e, and have higher Q-factor. For example, for normal incidence Q=109 for TE-mode and Q=58 for TM-mode. The reason is that TE-mode is concentrated predominantly in the smooth BIG sublayer (see Fig.2a,c) while the TM-mode penetrates into the nanopillars significantly. Thus, TM mode is more sensitive

to the perforated structure of the guiding layer and is scattered stronger than TE-mode so that $Q_{leak}^{TM} < Q_{leak}^{TE}$.

6. The TMPIE is interesting. The authors qualitatively commented its origin. I would strongly recommend making the explanation quantitative. What is the expected angle α from this material and structure, and how does this match with experiments?

One can estimate the angle α from the longitudinal magneto-optical effect which is also enhanced in these structures due to the eigenmode polarization variation but refers to the different configuration of the magnetic field: also in-plane, but parallel to the guided mode wavevector.

We have performed the simulations of this longitudinal polarization rotation effect at the angle of incidence 44° :

For the mode propagating at an angle $\varphi = 37^\circ$ to the incidence plane if $\theta = 44^\circ$ and polarization variation α roughly estimated as longitudinal polarization rotation shown in figure above. The amount of incident light energy converted to the mode is proportional to the square of projection of the incident \mathbf{E}^{inc} on the guided mode eigen polarization \mathbf{E}^{wg} . One may get the following estimation for the efficiency of light conversion to mode $\eta = \cos^2(\varphi \pm \alpha)$ and its variation:

$$\Delta\eta = \cos^2(\varphi + \alpha) - \cos^2(\varphi - \alpha) = \sin(2\varphi) \sin(2\alpha)$$

Therefore, the numerical simulations give $\alpha \approx 0.05^\circ$ tilt angle of the mode eigen polarization. It corresponds to the $\frac{\Delta\eta}{\eta} \sim 0.26\%$ variation of the coupling efficiency. This numerical estimation is very close to the experimentally observed transmittance variations $\Delta T/T \sim 0.2\%$ for these modes.

Also, from numerical simulations, more work can be done by simulating the transmittance spectrum and magneto-optical effect. This could be solved by

considering the tensor permittivity of BIG in finite element analysis or FDTD simulations. So that a quantitative compare with experiments can be observed.

We have added numerical simulations in Supplementary S4. Simulations were performed using the RCWA method.

The positions of the observed guided mode resonances show a very good correspondence with the numerical results. The numerically obtained values of magneto-optical modulation for both p- and s- polarizations also show a good correspondence with experimentally obtained values.

The magneto-optical response of the upper TE(0,1) mode can be considered mainly TMPIE, and the U-shape of the TMPIE resonance and its opposite sign for p- and s- polarization is clearly seen. For the other modes, the total magneto-optical response is a sum of the 'transversal' and the 'longitudinal' contributions, which have different origins (mode dispersion and polarization variation, correspondingly, as discussed in the manuscript). The exact value of these contributions may vary due to some structure imperfections. Thus, the shape of some resonances differs from the obtained in experiments, however, it is important that the absolute values of the observed magneto-optical modulation is nearly the same.

Transmittance and magneto-optical intensity modulation spectra (numerical simulations).

7. In the TMPIE section, for the modal coupling between TE and TM due to MI, would this effect compete with the birefringence of these two modes? Would the polarization show a beating pattern along the propagation direction? How does this effect relate to the free space coupling?

We agree that this question is really important as the polarization transformation lies in the root of the reported TMPIE.

There are two completely different cases for magnetized waveguides when light of a frequency ω incident at a fixed incidence angle θ could excite either (1) both TM and TE modes which collinearly propagate or (2) only one of the TM or TE modes.

Case 1) has been studied since 1980s (see, for example, a review Prokhorov A M, Smolenskii G A, Ageev A N *Sov. Phys. Usp.* 27 339–362 (1984)). There are many ways to implement it. For example, one may take a thick film to get relatively larger density of modes so that TE and TM propagation constants will be close to each other (e.g. films of 3-7 μm thickness considered by Prokhorov et al.). In this case the beating pattern of mode polarization was observed due to the simultaneous excitation of magnetically coupled TM- and TE-modes with slightly different wavenumbers.

However, our study focuses **on a very thin film where dispersion curves of the TE and TM modes are rather distant and only one of the modes could be excited by light of frequency ω incident at the fixed θ** (see Fig.3be – the transmittance spectra where TM and TE resonances are notably separated from each other). Therefore, here we deal with the case 2). (The exception is the points of the dispersion interception, such as at 5deg, 790nm – see Fig.3b,e, however the TM and TE modes are excited in the opposite +Ox and -Ox directions and do not interact with each other).

Let us analyze what happens in our case. The modes have linear eigen polarization in the non-magnetized case, for example, TM-mode contains E_x, E_z, H_y components of electromagnetic field. When we switch on the longitudinal external magnetic field **exact analytical solution of Maxwells' equation** gives the different polarization of the eigen mode: the **eigenmode** still has the same E_x, E_z, H_y components but acquires TE-like E_y, H_x, H_z components linear in magnetization. These components E_y, H_x, H_z are not associated with some other TE-mode existing in the structure and appear as the solution of the Maxwell's equations due to the magnetic field induced modification of constitutive equations. Therefore, quasi-TM mode appears. Its propagation constant remains the same as for TM-mode in the non-magnetic case. Thus, the ratio and phase shift between all of the eigenmode electromagnetic field components remain the same during mode propagation. If such eigenmode is excited, it propagates without a polarization rotation (in contrast to the beating polarization pattern in the case-1). However, the ratio of the TE/TM

components is rather small in this case, approximately $1e-3$ ($\tan^{-1}(E_y/E_x) = 0.05$ deg) (compared to the case-1 where 100% transformation is possible).

We give the reference [51,52] to our previous works (page 8), where analytical solution of Maxwell's equations for plasmonic waveguides is provided in the form of quasi-TM and quasi-TE modes. Similar expressions are valid also for the all-dielectric waveguide.

We also agree that some terms used for the description of this process, such as 'polarization rotation' instead of 'modification', were confusing and significantly revised this part of the manuscript to make it more clear (see page 8).

8. There are some typos, please consider revise.

We have revised the manuscript and corrected the typos.

Reviewer #3 (Remarks to the Author):

This is a very good manuscript reporting timely results. I totally agree with the claim in the manuscript that 'it might be advantageous to modulate both orthogonal linear polarizations, i.e. p- and s-polarizations of light'. The authors are also right to claim that this might have important practical applications. Overall, this manuscript is a significant contribution to the rapidly developing field of all-dielectric magneto-photonics that goes hand-in-hand with a large and growing body of research on all-dielectric nanophotonics and perhaps on metamaterials and metasurfaces. Although I do not always agree with the theoretical interpretation by the authors, they published many works explaining their point of view in the past and this appears to be well-accepted by the community. Therefore, I would recommend this present manuscript for publication with minor suggestions (see below).

We would like to thank the Reviewer-3 for the supportive comments about our manuscript and importance of the results.

1)I am not really surprised to see that nanostructure-assisted magneto-optical effects in transmission can be observed not only for p-polarisation but also for s-polarization of the incoming light. For instance, for simplicity let us consider the transmission of light through subwavelength apertures in metal films. (Similar effects also occur in dielectric nanostructures under appropriate conditions.) Let us also focus on the relevant phenomenon of extraordinary optical transmission in

periodic arrays. It is known that surface electromagnetic modes play a key role in the emergence of the resonant transmission. See, e.g., F. J. Garcia-Vidal, L. Martin-Moreno, T. W. Ebbesen, and L. Kuipers, *Rev. Mod. Phys.* **82**, 729 (2010). Usually, such surface electromagnetic modes are excited using p-polarised light; however, they can also be excited using s-polarised light. The review paper above cites several relevant works. Can the authors comment on this and extend their discussion if appropriate?

Thank you for raising this issue. We quite agree that the nanostructured materials can provide the various types of the resonance for both p- and s- polarizations of the incident light. We also agree that some of the magneto-optical effects, such as longitudinal and polar Kerr and Faraday effects that are responsible *for polarization rotation* were observed in both polarizations and were reported to be enhanced in different types of nanostructures. However, we would like to point out, that, to our knowledge, this is *the first experimental demonstration of the magneto-optical intensity modulation of s-polarized light* in transversal configuration. Various types of nanostructures, with EOT, plasmonic, LSP, Mie, etc. resonances reported up to nowadays, modulate only the transmittance of the p-polarized light. This limitation comes from a magneto-optics of the smooth magnetic films where transversal magnetization affects only the constitutive equations for the p-pol. light and leaves s-pol. light undisturbed.

We have added the references on page 2 to the review about EOT nanostructures [24] and recent works on magneto-optics [25,26] of such EOT nanostructures. Similar to the other plasmonic, dielectric structures, *the magneto-optical light intensity modulation was observed only for p-polarization of light in EOT nanostructures.*

2)The term ‘metasurface’ is very popular. However, too many authors abuse it, which I suspect might be also the case of this work. Please justify why the proposed structure is a metasurface.

We quite agree that there could be a few ways to refer our structure, e.g. a ‘planar 2D photonic crystal’, ‘thin waveguide with extended coupling grating’, ‘2D nanopillar array’ etc. But, to our opinion, such terms do not fully describe the essence of the proposed system and its optical response.

Let us cite the definition of metasurface given in a recent review ‘Optical Metasurfaces: Progress and Applications’ (*Ann. Rev. of Materials Res.*, 2018 48:1, 279): ‘metasurface is an artificial nanostructured interface that has subwavelength thickness and that manipulates light by spatially arranged meta-atoms—fundamental building blocks of the metasurface’.

It fully characterizes our subwavelength BIG film with nanopillars as ‘meta-atoms’ the spatial arrangement of which is responsible for the observed novel magneto-optical effect.

Let us underline the meaning of the prefix ‘meta-’ as ‘beyond’: the demonstrated magneto-optical modulation of s-polarized light is prohibited in magnetic smooth films and bulk crystals and is due to the specially designed nanostructuring.

Also, there are other recent works in which similar structures based on subwavelength gratings with guided modes are referred as metasurfaces, such as:

1. Han, S., Rybin, M. V., Pitchappa, P., Srivastava, Y. K., Kivshar, Y. S., & Singh, R. (2020). Guided-Mode Resonances in All-Dielectric Terahertz Metasurfaces. *Advanced Optical Materials*, 8(3), 1900959.
2. Pidgayko, D., Sinev, I., Permyakov, D., Sychev, S., Heyroth, F., Rutckaia, V., & Samusev, A. (2018). Direct imaging of isofrequency contours of guided modes in extremely anisotropic all-dielectric metasurface. *ACS Photonics*, 6(2), 510-515.
3. Yildirim, D. U., Ghobadi, A., Soydan, M. C., Gokbayrak, M., Toprak, A., Butun, B., & Ozbay, E. (2019). Colorimetric and Near-Absolute Polarization-Insensitive Refractive-Index Sensing in All-Dielectric Guided-Mode Resonance Based Metasurface. *The Journal of Physical Chemistry C*, 123(31), 19125-19134.

etc.

3) Regarding the ‘Transverse magnetophotonic intensity effect’. Can the authors justify the need to call this a new effect? As I said above, this seems to be the same effect but for another light polarisation.

Magneto-optical effects differ from each other in the configuration under which they are observed, their origin and provided impact on optical radiation. Let us summarize the known magneto-optical effects observed for the in-plane magnetization to reveal the differences:

	Longitudinal Kerr effect (LMOKE)	Transverse Kerr effect (TMOKE)	Voigt effect	Longitudinal magnetophotonic intensity effect [NatComms 4, 2128 (2013)]	Reported here transverse magnetophotonic intensity effect
Type of light modulation	polarization	intensity	polarization	intensity	intensity
Parity	odd	odd	even	even	odd
Incident light polarization	both	P	any except pure p&s	both	both

light incidence	oblique	oblique	any	normal	oblique
Origin	boundary conditions	boundary conditions	magnetic birefringence	TM-TE mode coupling	mode polarization
In smooth films	allowed	allowed	allowed	prohibited	prohibited

We believe that this table justifies that similar to the first 4 effects which are commonly-accepted to be different, the reported TMPIE is also different from all of them.

REVIEWERS' COMMENTS

Reviewer #1 (Remarks to the Author):

all my comments have been correctly addressed.
This article can be published.
thanks

Reviewer #2 (Remarks to the Author):

The revised manuscript answered all my questions and provided a much clearer picture of the new phenomena and physics found in this work. This is a timely and impressive work toward highly efficient all dielectric MO metasurfaces. I support the publication of this work in Nature Communications.

Reviewer #3 (Remarks to the Author):

The authors addressed all my comments, concerns and suggestion in great detail. The revised manuscript was also significantly improved by addressing the comments by the other two Referees. In my opinion, the revised work meets the high standards of the journal. I recommend it for publication.

Response to reviewers` comments

Reviewer #1 (Remarks to the Author):

all my comments have been correctly adressed.
This article can be published.
thanks

We thank the Reviewer #1 for the evaluation of the work and the important comments.

Reviewer #2 (Remarks to the Author):

The revised manuscript answered all my questions and provided a much clearer picture of the new phenomena and physics found in this work. This is a timely and impressive work toward highly efficient all dielectric MO metasurfaces. I support the publication of this work in Nature Communications.

We thank the Reviewer #2 for the high evaluation of the work and the important comments.

Reviewer #3 (Remarks to the Author):

The authors addressed all my comments, concerns and suggestion in great detail. The revised manuscript was also significantly improved by addressing the comments by the other two Referees. In my opinion, the revised work meets the high standards of the journal. I recommend it for publication.

We thank the Reviewer #3 for the high evaluation of the work and previously highlighting the aspects the were needed to be clarified.